# An Atheistic Argument from Naturalistic Explanations of Religious Belief: A Preliminary Reply to Robert Nola

**Kai-man Kwan**

Department of Religion and Philosophy, Hong Kong Baptist University, 224 Waterloo Road, Kowloon, Hong Kong; kmkwan@hkbu.edu.hk

**Abstract:** Robert Nola has recently defended an argument against the existence of God on the basis of naturalistic explanations of religious belief. I will critically evaluate his argument in this paper. Nola's argument takes the form of an inference to the best explanation: since the naturalistic stance offers a better explanation of religious belief relative to the theistic explanation, the ontology of God(s) is eliminated. I rebut Nola's major assumption that naturalistic explanations and theistic explanations of religion are incompatible. I go on to criticize Nola's proposed naturalistic explanations: Freudianism, a Hypersensitive Agency Detection Device, and a Moralising Mind-Policing God. I find these inadequate as actual explanations of religious belief. Even if they are correct, they will not show that theism is false. So Nola's argument fails to convince.

**Keywords:** Robert Nola; naturalistic explanation of religious belief; cognitive science of religion; hypersensitive agency detection device

## 1. Introduction

In contemporary philosophy, theists and atheists continue to argue for or against the existence of God. Robert Nola has recently defended an argument against the existence of God on the basis of naturalistic explanations of religious belief (Nola 2013, 2018). A naturalistic explanation is a non-epistemic explanation of religious beliefs that does not assume the existence of the object of that belief, i.e., God. There are four main types of naturalistic explanations: psychological, sociological, neurophysiological, or evolutionary explanations (this is often coupled with Cognitive Science of Religion, CSR). Nola's favored naturalistic explanations are psychological and evolutionary (CSR, in particular) and appeals entirely to natural processes that are apparently unrelated to the quest for truth.

In fact, Nola has offered two arguments against theism, which are related yet distinct. He explicitly says that "there are two targets of debunking arguments"—"an act such as x's believing that God exists, and the putative state of affairs expressed by the proposition that God exists" (Nola 2013, p. 164). The first argument directly argues against the existence of God and argues for the conclusion of atheism from the superiority of naturalistic explanations of religious beliefs over supernaturalistic explanations of religious belief (call this kind of argument SNE). This is the argument that will be examined in some detail in this paper. The second is a debunking argument against religious beliefs from the alleged unreliability of the processes or mechanisms, which lead to the formation of religious beliefs (call this this kind of argument DRB). In fact, various kinds of debunking argument can be offered against science, morality, or religion.[1] In particular, the debunking arguments against religious beliefs on the basis of CSR have been hotly debated among philosophers and cognitive scientists of religion in recent years. The idea that such a debunking argument is available is espoused in Dennett's (2006) *Breaking the Spell*, and then defended by Liz Goodnick (2016) and Matthew Braddock (2016), among others.[2] There have been numerous responses to different forms of DRB, which can start with the falsity of god-beliefs, the diversity of god-beliefs, or the unreliability/insensitivity of the processes that lead to the

formation of religious beliefs. The major respondents include Michael Murray, Kelly James Clark, Justin Barrett, Joshua Thurow, Aku Visala, and Hans van Eyghen.[3] This debate between the defenders and critics of DRB features prominently in some new books about CSR, such as those by Trigg and Barrett (2014), De Cruz and Nichols (2016), and Van Eyghen et al. (2018).

In these recent works, various forms of DRB have been examined rigorously, and Nola's version has not escaped attention.[4] Responses to Nola's DRB include Van Eyghen (2016), Van Eyghen (2019), and Launonen (2021), and, in any case, discussions of DRB in general have been extensive in the current literature. In contrast, there seems to be no detailed response to Nola's SNE yet. Hans van Eyghen notices that "Nola (2013) presents two arguments in his article" (Van Eyghen 2016, p. 980, n. 8) and that Nola tries to use an inference to the best explanation (IBE) to "eliminate folk ontology of gods" (Van Eyghen 2016, p. 971). However, the purpose of that article is mainly to explain Van Eyghen's classification of explaining away arguments in CSR into two types: one type is an argument for incompatibility, and another type is an argument for superfluity. Since van Eyghen is using Nola mainly as an example to illustrate his classification, Nola is, in fact, not quoted a lot and there is no detailed reply to Nola beyond a schematic response. Walter Scott Stepanenko (2021) has provided a more substantial response to Nola's SNE. His basic position is that "Nola's argument for the superiority of naturalistic explanations over religious explanations is irresponsive to first-order evidence", and, hence, "the argument . . . rests entirely on methodological appeals" (Stepanenko 2021, p. 7). However, for Stepanenko, since "evaluative criteria are derived from methodological commitments", which are worldview-dependent, it will generate a "problem of evaluative incommensurability" (Stepanenko 2021, p. 9).[5]

In this paper, I intend to offer a more detailed response to Nola's SNE. I will argue that naturalistic explanations can be compatible with theism and that the naturalistic explanations of religious beliefs proposed by Nola are far from convincing. However, I would not adopt Stepanenko's strategy to appeal to incommensurability. Although I fully recognize the trickiness of arguments between different worldviews because of the difficulty of avoiding circular reasoning, I do believe that there are rough common criteria (albeit not completely precise) that can be used to compare theism and naturalism.[6] Nola has further elaborated his SNE later in his 2018 article, but a full response to that paper would require lengthy and in-depth discussions about what are the correct criteria to evaluate explanations in commonsense reasoning, science, and religion and how to interpret them. This cannot be done here, and, hence, this paper will focus mainly on Nola (2013). I hope to deal with the methodological issues more fully in another paper.

As pointed out above, there seems to be many more responses to Nola's DRB than his SNE, but I do believe that Nola's argument for atheism deserves a more detailed response. His SNE seems to presuppose the logical incompatibility between naturalistic explanations of religious beliefs and their theistic explanations, and, in fact, this is not infrequently criticized by both theists and atheists (see discussions below). However, at the same time, quite a few CSR scholars and critics of religion seem to suggest or imply that there is *some kind of* incompatibility between naturalistic explanations of religious belief and their theistic explanations, even if this does not amount to logical incompatibility, and this general inclination seems to be shared by many educated people. It seems that this is because the availability of naturalistic explanations of religious belief fits quite well with the hermeneutics of suspicion, one characteristic mentality of the modern period. For example, Nietzsche, one master of suspicion, thinks that an insight into the origin of belief in God would render "a counter-proof that there is no God . . . superfluous" (quoted in Schupbach 2016, p. 256). Perhaps Nietzsche is mistaken here, and the alleged incompatibility is an illusion. However, even if this is the case, this is a widely shared illusion which needs to be dispelled more thoroughly. That is why I think a more extended treatment of Nola's SNE is justified. To do that, we also need to interpret "incompatibility" more broadly as "hypothesis competition", which includes logical incompatibility but also more.

Let me now introduce Nola's arguments.

## 2. From Naturalistic Explanations of Religious Beliefs to Denial of Theism

Nola proposes an argument against belief in God on the basis of naturalistic explanations of religious beliefs. This takes the form of an inference to the best explanation (IBE): "The various scientific accounts recently developed of the causes of religious beliefs have gained considerable explanatory efficacy . . . the naturalistic stance offers the better explanation" relative to the "folk explanatory hypothesis, F, which supposes that ultimately the divinities . . . do exist and have a casual role in bringing about" religious belief. So according to IBE, "it can be inferred that it is far more reasonable to believe the naturalistic rather than the supernaturalistic explanation" (Nola 2013, p. 162). Nola also claims that, "*In accepting this conclusion* F is thereby rejected and its ontology of God(s) is eliminated . . . In this manner IBE provides one kind of debunking argument aimed directly at the claim that Gods exist" (Nola 2013, p. 167, italics mine).

One of Nola's major assumptions is that naturalistic explanations and folk explanations (or theistic explanations in my terms) of religion are incompatible. For him, naturalistic explanations only appeal to "items and processes (physical, mental or social) found in the space-time framework that form a casual nexus and are well-attested postulations of science". So they "rule out any appeal to supernatural entities". In contrast, "a common 'folk' explanation of religious beliefs is in terms of the existence and causal efficacy of supernatural agents" (Nola 2013, p. 165). The "naturalistic and 'folk' explanations are incompatible, particularly at the level of the ontologies they respectively propose" (Nola 2013, p. 166).

Having laid down the framework, Nola goes on to propose three promising candidates for the naturalistic explanation:

### 2.1. Freudianism

"We need not accept all of Freud's dubious theory of oedipal complexes and the panoply of deeply unconscious wishes. However some of the claims Freud makes about wish-fulfilment, when disconnected from his psychoanalytic theories, are initially plausible and open to test" (Nola 2013, p. 170). However, "wish-fulfillment . . . is an unreliable BFP [belief-forming process] . . . there is no reason for the Freudian wish-fulfillment BFP to be oriented towards the truth at all; its aim, we might say, is not to uncover truths about the world but to provide beliefs that are comforting and enable us to cope, whatever their truth-value". (Nola 2013, p. 171)

### 2.2. Hypersensitive Agency Detection Device (HADD)

"Error Management Theory proposes that natural selection is to be understood to favour those errors which are less costly, and not to favour those errors which are more costly, particularly those that are fatal", and it, in the end, produced an "error-prone ADD. It will make Type I errors in giving predator alerts when there is no predator; the cost here is that of being unnecessarily alert when one might do other things that are to one's advantage" (Nola 2013, p. 174). Moreover, "various aspects of human cognition mentioned, such as HADD, ToM and MCI, can come together in the human attribution of agency to nature" (Nola 2013, p. 176). ToM (Theory of Mind) and MCI (Minimally Counterintuitive beliefs) are also mechanisms postulated by CSR, the details of which we will not go into here.

Hence, "HADD will not be a reliable detector in the sense of delivering truths most of the time when it operates" (Nola 2013, p. 175), because it is liable to be "off-track". "The very idea of HADD having evolved requires that it be hypersensitive, and this contributes to its unreliability . . . Those within evolutionary psychology who advocate HADD as a cause of religious beliefs are on a path towards a debunking of those beliefs" (Nola 2013, p. 177).

*2.3. Moralising Mind-Policing God ("MMP-God")*

Nola also appeals to a theory in evolutionary psychology, which directs us to the problem of explaining "how cooperation is maintained in the face of obvious defection by cheaters and freeloaders without the kinds of controls that can be found in modern secular societies (e.g., police, courts . . . ) The suggestion is that religion plays a large role here with its promises of rewards in the next world; but more important for this-world selection processes are the punishments that can be meted out to defectors by devotees of an MMP-God" (Nola 2013, pp. 181–82).

So Nola believes that natural selection would favor a "belief in a Moralising Mind-Policing God ("MMP-God") who knows all our inner beliefs and desires", because "beliefs in an MMP-God will be a potent factor in preventing us from being wicked, or of cheating our fellow members in society" (Nola 2013, p. 180). Hence, "evolution provides a cause which explains acts of believing in a MMP-God . . . there is no reason to suppose that the causal mechanism is reliable for the truth of the contained belief . . . Rather it produces beliefs which enhance cooperation and survival" (Nola 2013, p. 182).

Nola believes that once we feed these good scientific theories of religion into his IBE, the belief in God will be defeated. Before evaluating his argument, we should note that there is some tension between the second and third naturalistic explanations proposed by Nola. The MMP-God theory provides a selectionist explanation that regards religion itself as an *adaptation*. The appeal to HADD belongs to contemporary mainstream CSR, but "most scholars working in the CSR reject selectionist explanations in favour of a structural constraints-based account of the evolution of religion". So "religion is an incidental *byproduct* of domain-specific cognitive adaptation that underwrite and constrain the universe of religious representations and their associated behaviours" (Powell and Clarke 2012, p. 459, italics mine). I will later point out that the tension can be removed by allowing multiple kinds of explanations to co-exist in CSR.[7]

## 3. Naturalistic Explanations Are Not Incompatible with Theistic Explanations

Nola believes that naturalistic explanations and folk explanations of religion are incompatible because naturalistic explanations "rule out any appeal to supernatural entities". First, we need to point out that to "rule out any *appeal to* supernatural entities" is a necessary rule of game, or methodological commitment, for those who propose naturalistic explanations—they simply would not *adopt* any supernatural hypothesis in their *naturalistic* explanations for obvious reasons. However, this does not mean we have to "*rule out* supernatural entities" in the ontological sense. For example, a sociologist typically does not appeal to psychological processes when they provide a sociological explanation of a phenomenon such as suicide; for example, when they argue that the cultural ethos of a society contributes to a higher suicide rate in that society relative to another society that has a lower suicide rate. However, this sociological explanation obviously does not rule out psychological explanations of suicide as possible truths. In fact, given a shared cultural ethos, the sociological explanation alone cannot explain why one individual in that society committed suicide while another individual in the same society did not.

Naturalism is indeed an ontology excluding the supernatural, and Nola is a supporter of *scientific naturalism*, and points out that "such a naturalism not only comes into conflict with the framework of "folk" explanation of religious beliefs, but if adopted eliminates that framework" (Nola 2013, p. 178). However, it is difficult to understand why naturalistic explanations "rule out any appeal to supernatural entities", unless we identify naturalistic explanations with naturalism. However, naturalistic *explanations* in themselves do not presuppose a comprehensive naturalistic *ontology*, because they only propose that natural entities or processes are sufficient to explain the rise or persistence of religious beliefs. This does not entail that natural entities or processes are all that exist—and only this amounts to a position of naturalism. At most, supporters of the naturalistic explanation may adopt a rule of *methodological* naturalism, which should be distinguished from *metaphysical* naturalism.

So Nola's major assumption of the incompatibility of a naturalistic explanation and a "folk" explanation seems false.

Nola may say that the alleged incompatibility is an implication of IBE: naturalistic explanation, as "the better explainer . . . then eliminates the folk ontology of God(s)" (Nola 2013, p. 182). However, it is a misunderstanding. Take a detective case as an example. A detective has evidence to show that either suspect A or suspect B is the most plausible candidate for being the murderer. Suppose further evidence shows that the "suspect A hypothesis" provides a much better explanation than "the suspect B hypothesis". Should we not then eliminate the "suspect B hypothesis" and accept A as the murderer? Normally this is the case, but this is only because the detective has assumed from the beginning that usually there is *only* one murderer. This assumption may be justified by past statistical evidence and the Principle of Simplicity. The hypothesis that there are two or more murderers would unnecessarily complicate the case and needs to explain matters such as how they can collaborate, etc. However, suppose, in the above case, that the suspect A theory still leaves many loose ends untied, and the detective finds out that the "both murderers hypothesis", in fact, is the best overall explanation of *all* the evidence. In this case, an IBE may reasonably arrive at the conclusion that both A and B are murderers.

Of course, when it is not likely to have such collaboration, the detective may only consider two mutually exclusive hypotheses:

**Theory 1.** *A alone is the murderer.*

**Theory 2.** *B alone is the murderer.*

In this case, if Theory 1 is a better explanation than Theory 2, and there are no other plausible theories, then the IBE will properly conclude that Theory 1 is true. However, if the prior assumption that it is not likely for two or more persons to collaborate in murder is mistaken, the detective's IBE needs to consider more hypotheses than just Theory 1 and Theory 2.

In the case of testing comprehensive theories of natural law (such as T1 and T2) which purport to explain all empirical phenomena, by the nature of the case T1 and T2 are mutually exclusive (unless one entails the other, which is rare). So if T1 wins in an IBE, T2 will be eliminated and vice versa. However, in the case of explaining the spread of a disease, suppose several explanations, E1, E2, and E3, are proffered. Even if E1, for example, the new virus hypothesis, is by far a much better explanation than the others, normally E2 and E3 will not thereby be eliminated. This is because we know that many factors that contribute to the spread of a disease (e.g., poor sanitation conditions and a backward state of medicine) need not be mutually exclusive. Each factor may only provide a *partial* explanation.

Nola may also adduce an explanatory argument for naturalism by appealing to the Principle of Simplicity. He may contend that since naturalistic explanation is the better explanation, adding theism on top of that, though not logically impossible, would violate the Principle of Simplicity. So it is more reasonable to accept naturalism and reject theism. This is a serious argument, but there are two problems.

First, different scholars have different understandings of simplicity. For Robert Segal, "a naturalistic account is simpler than a religious one" because "a religious account postulates an entity *in addition to*, not *in place of*, all the natural ones already postulated . . . By contrast, naturalistic accounts either use already postulated entities and processes like the unconscious and projection or at most propose new ones comparable with the ones already postulated" (Segal 1992, p. 71). Here, Segal understands simplicity mainly as ontological economy or parsimony, which aims at reducing the number of types of entity postulated in a worldview. However, Swinburne contends that the introduction of God simplifies the whole worldview by using only *one* Entity to explain and unify various aspects of the world that are hardly explicable or expected on naturalism.[8] It seems that the principle of simplicity itself is far from simple, and there are ongoing complicated debates about this concept and the application of this principle to assessment of theories in science and

philosophy.[9] Of course the debate deserves a fuller treatment and cannot be resolved here. Here, I only want to point out that such an explanatory argument for naturalism is by no means uncontroversial.[10]

Second, both naturalism and theism are worldviews that purport to be able to explain everything, at least in principle. So explanatory arguments for worldviews need to consider all the data, and not just a piece of datum. Even if naturalism could provide an overall better explanation of the phenomena of religious beliefs than theism does, the conclusion is only that the phenomena of religious beliefs, as part of the data, can raise the probability of naturalism relative to theism. As such, it may be regarded as part of the cumulative case for naturalism, but it is still too early to declare the final victory of naturalism. So the implications of Nola's SNE, even if successful, are limited. The theists may still argue that on the whole theism is the better explanation because its explanations of phenomena such as fine-tuning, moral laws, and so on are superior to those provided by naturalism. Both theists and atheists have tried to use this kind of cumulative explanatory argument to vindicate either theism (see Swinburne 2004; Forrest 1996), or atheism (see Beyer 2007). Of course, we cannot demand Nola to produce such a wide-ranging argument in the space of several articles. For the time being, the theists, as I argue in this preliminary response, seems to be justified in saying that Nola's SNE, an IBE restricted to the data of religious beliefs, is far from conclusive. To produce a more convincing and fuller reply, the theists also need to further substantiate their case, but this is beyond the scope of this paper.

## 4. Types of Religious Explanation and Levels of Explanation

So even if it is far more reasonable to believe the naturalistic rather than the supernaturalistic explanation of religious beliefs, it does not follow that it is unreasonable to believe in God. Nola's target is "a common "folk" explanation of religious beliefs . . . in terms of the existence and causal efficacy of supernatural agents" (Nola 2013, p. 165). He also clearly asserts that "in the case of explaining religious beliefs the naturalistic and "folk" explanations are *incompatible*, particularly at the level of the ontologies" (Nola 2013, p. 166, italics mine). However, among these folk explanations, we need to distinguish crude or direct supernaturalistic explanations from sophisticated theistic explanations of some event E. A crude supernaturalistic explanation of E would appeal to *direct, unmediated, or sometimes miraculous* actions of God, whereas a sophisticated theistic explanation would allow the divine causality of E to be *mediated by secondary causes* including "items and processes found in the space-time framework". So the validity of a naturalistic explanation would, at most, defeat crude supernaturalistic explanations but is perfectly compatible with a sophisticated theistic (or theological) explanation. As Jonathan Jong recognizes, "A successful naturalistic explanation of a particular religious belief system, R, might count against R if it insists on its own supernatural origins" (Jong 2013, p. 526), but not otherwise.[11] However, a sophisticated theist may fully admit the truth of the naturalistic explanation of their religious belief but insist that the processes or mechanisms postulated in that naturalistic explanation (evolutionary history of human beings, HADD, etc.) are themselves created by God in order to bring about religious beliefs. So even if the premises of Nola's IBE were true, it does not really show that theism is false.[12]

In general, naturalistic and theological explanations of phenomena may conflict only when both occur at the same level of explanation. While crude supernaturalistic explanations may compete with naturalistic explanations at the same level, theistic explanations need not and typically do not. We can illustrate this point by looking at various naturalistic explanations themselves in CSR. Some research on CSR "has focused on our cognitive architectures and processes", but this "level of analysis . . . leaves unanswered questions about how our minds got to be that way". In contrast, the "work on anthropomorphism and teleo-functional reasoning has lent itself to developmental . . . and evolutionary analyses . . . In none of these cases do the different kinds of explanations—cognitive, developmental, and evolutionary—compete, occurring as they do at different levels". For example, "the

fact that there is a successful evolutionary theory of religious phenomena would hardly count against developmental or cognitive theories" (Jong 2013, p. 527).

Interestingly, Nola does not appeal to one naturalistic explanation alone, but three. So are his three naturalistic explanations mutually incompatible? Will the success of HADD rule out the appeal to Freudianism? Nola's own practice suggests a negative answer to these questions. I have pointed out that there is some tension between Nola's second and third naturalistic explanations. However, strictly speaking, the adaptionist MMP-God theory does not conflict with the appeal to HADD. The tension exists between two research paradigms mainly because each paradigm may have methodological worries about the other (Powell and Clarke 2012). So if Nola can insist that his proposed naturalistic explanations do not "rule our or eliminate" one another, why does he insist that those three naturalistic explanations would eliminate the sophisticated theistic explanation? This is especially so because, concerning the explaining of religion, in fact, there are several kinds of explanation.

For example, we can distinguish the origin explanation (the very origin of religion in human history, or phylogeny), from the formation of religious belief in a certain individual (the ontogeny of religious belief).[13] Freudianism is supposed to give answers to questions about both phylogeny (primal parricide) and ontogeny (wish fulfillment). Similarly, in CSR "there are evolutionary causes of developmental facts, and developmental causes of facts about cognitive architecture and process and, finally, cognitive causes of religious phenomena. In this case, cognitive explanations of religion are proximate explanations, whereas evolutionary explanations of religion are ultimate explanations". So "multiple levels of analysis are far from foreign in CSR" (Jong 2013, p. 527). Just as "cognitive, developmental and evolutionary explanations of religion do not compete with one another, theological explanations do not compete with either of them insofar as they occur at a still higher level of explanation than do evolutionary theories" (Jong 2013, p. 528). This can be called the Metaphysically Ultimate level.

Theists explain the existence of individuals living today by divine creation, but this explanation does not conflict with the scientific explanation of human existence in terms of the science of reproduction. While the latter provides the proximate explanation of our existence, the Creator who established all these mechanisms in the beginning provides the Metaphysically Ultimate explanation. Both can co-exist happily.[14] Similarly, proximate explanations of religious belief in terms of naturalistic processes and Metaphysically Ultimate explanation of religious belief in terms of the Creator's action can co-exist happily. Since one crucial premise of Nola's IBE is false, his argument cannot really go through.

*Explanatory Pluralism and Competing Explanations*

The attractiveness of SNE seems to be supported by a widespread belief that the only legitimate kind of explanation is a scientific explanation with reference to the properties of the fundamental physical constituents of things and the natural laws governing their behaviour. If there are other legitimate explanations in a higher level of science such as psychology or sociology, they must be reducible in principle to this kind of naturalistic or physicalist explanation. In the heyday of logical empiricism or positivism, this kind of position, which is called Explanatory Fundamentalism (EF) by Visala, was quite popular. Nowadays, although EF is still influential, its popularity has waned somewhat, and it is challenged by advocates of Explanatory Pluralism (EP). For example, Visala says, "instead of adhering to some robust naturalist metaphysics, we should adopt a more pragmatic approach to explaining religion. Causal explanations do indeed float relatively free of the material constituents of their targets and explainers. This makes the EF commitment to reduction of all causal explanations of culture to its lower-level components . . . dubious" (Visala 2014, p. 57).[15]

There are different ways to challenge EF. One way is to adopt a non-naturalist ontology and to argue that explanations with reference to non-physicalist entities such as mind, objective meaning, or moral facts are at least possible. The other way is to combine a

broadly naturalistic worldview with emergentism, which posits emergent levels of reality not reducible to their physical constituents (Clayton 2004). The general idea is that "the world is complexly layered at numerous measurement scales, admitting of diverse patterns and principles that may look very different across these scales" (Dale et al. 2009, p. 3).

However, in challenging EF, Visala wants to adopt a more pragmatic approach: what is a proper explanation is relative to our explanatory interests or contrast spaces. In a widely discussed example of a bank robber, Willie Sutton, who was asked by the chaplain, "Why did you rob the banks?" Sutton replied, "That's where the money is". For Sutton, the explanatory interest is "why rob banks rather than other places" but for the chaplain, the explanatory interest is "why rob at all" (see Visala 2014, p. 67ff). Once we admit this relativity of proper explanations, EF would look overly restrictive. However, at the same time, without specifying the contrast space, EF is at the same time excessively rich. For example, when we seek the explanation of a car crash, the relevant physicalist explanations with reference to the physical constituents of the car and other things involved are just too many. So one major problem of EF is that it is "unable to identify those causal relationships that are relevant for a given phenomenon" (Visala 2014, p. 63).

For Visala, EP is built on the idea that explanations "are answers to what-if-things-had-been-different questions. As such, their explanatory frames float relatively free of the physical constituents of the target and the explainer" (Visala 2014, p. 69). Therefore, "we should not commit CSR to some general metaphysical theory of what causes of religion qualify as proper causes in the study of religion. Inquiry into religious phenomena can proceed in a piecemeal fashion, more 'locally' than 'globally', and it can be based on the usefulness of different kinds of explanations rather than *a priori* commitment" (Visala 2014, p. 72).

I think EP is a plausible alternative to EF, and it deserves more support.[16] Its more pragmatic approach helps to defuse the idea that only naturalistic explanations are legitimate explanations. It opens up our mind to the possibilities that naturalistic explanations can be combined or at least co-exist with other kinds of explanations such as mental, teleological or personal explanations. In this way, it weakens Nola's SNE. However, EP's major concern seems to be the defense of the partial autonomy of various analytic levels in different kinds of scientific disciplines. Although it argues against explanatory fundamentalism or hyper-reductionism, it does not explicitly deal with the tension between the naturalistic worldview and the supernatural worldview at the metaphysical level. For example, McCauley and Lawson want to assure the scholars of religious studies that they can "Stop Worrying about Reductionism" (heading of chapter one of McCauley and Lawson 2017). Their major conclusion is that "Scientific revolutions and the theoretical and ontological eliminations they underwrite occur between successive theories in some science, not between theories operating at different levels of analysis" (McCauley and Lawson 2017, p. 17). The implication is that despite advances in neuroscience, molecular biology, and so on, we do not need to worry that theories in religious studies would in the end be eliminated. However, this stops short of affirming the legitimacy of appealing to the actions of God as explanations. Visala (2014) makes no such affirmation either. So EP alone is not a sufficient reply to Nola's SNE.

So a naturalist may accept McCauley's EP but then insists that we stop at the physicalist level of explanation and must not go further. However, in line with EP's spirit of avoiding a priori metaphysical commitment, we can develop a pragmatic kind of EP into a more robust kind of EP that accepts the prima facie legitimacy of all kinds of questions, including metaphysical questions such as why there is something rather than nothing, why there are some natural laws rather than a chaos, and so on. Here I find Swinburne's extensive discussions of explanation and defense of the legitimacy of personal explanation (god-explanation is a kind of personal explanation) helpful (Swinburne 2004). In his defense of the design argument, he eschews appeals to direct divine interventions and accepts the completeness of scientific explanations. However, then he goes on to ask, "why there are such natural laws which allow for the existence of a stable temporal order which in the

end allows the emergence of living things and minds in our universe?" (These are also what if things had been different questions, in Visala's terms.) Science takes the existence of natural laws as its fundamental presupposition and goes on to explain everything else. So it cannot provide explanation of why such natural laws exist. However, theism can, and, in explaining the existence of natural laws with reference to God's intentions and actions, the theistic worldview can unify scientific explanation and personal explanation. This seems to be an explanatory virtue of the theistic worldview (Swinburne 1968). Of course, this involves complicated issues that need more thorough discussions. However, I think enough has been said to indicate that it is by no means easy for defenders of SNE to argue that the naturalistic explanations of religious belief in themselves logically exclude their theistic explanations.

However, defenders of SNE may retreat to a weaker version, which says that although there is no *logical* incompatibility between naturalistic explanations of religious belief and their theistic explanations, naturalistic explanations of religious belief would tend to disconfirm or undermine to some extent their theistic explanations. Schupbach and Glass point out that hypothesis competition "is a matter of degree" because "hypotheses that are not strictly speaking mutually exclusive may still do much to directly rebut each other" (Schupbach and Glass 2017, p. 811). They may compete indirectly via some body of evidence when "adopting either hypothesis undermines the support that the relevant body of evidence provides for the other" (Schupbach and Glass 2017, p. 812). In another paper, this kind of indirect explanatory competition is explained in this way: "distinct potential explanations compete epistemically with respect to their common explanandum if, upon accepting one of these, the other no longer retains its explanatory power" (Schupbach 2016, p. 259).

So let us consider whether naturalistic explanations of religious beliefs compete directly or indirectly with their theistic explanations, even if they are not logically incompatible. (Logical incompatibility would entail the maximum degree of direct competition.) The key question is whether the existence of naturalistic explanations of religious beliefs would render the existence of God or theistic explanations of religious beliefs unlikely, even if not impossible. Again, this seems to depend on the kind of theism we are considering. For a theism that requires a lot of divine interventions in the natural order in general, or a theism that requires religious beliefs to be directly produced by God, some kind of probabilistic direct competition does seem to occur. However, for a theism which conceives of a Creator God as working mainly through secondary causes, there is no such direct competition, probabilistic or otherwise. So if naturalistic explanations are well-established, they would not undermine theism as such, but may help to disconfirm versions of crude supernaturalistic theism.

What about indirect competition then? It initially seems that a case can be made here because naturalistic explanations of religious beliefs, and their theistic explanations have religious belief as their common explanandum. Moreover, if we accept naturalistic explanations of religious beliefs, apparently theistic explanations would be rendered redundant, and, hence, they no longer retain their explanatory power. This seems to satisfy Schupbach's (2016) condition for indirect competition. However, Schupbach also explicitly lays down prerequisite conditions for this kind of competition: "wholly distinct types of potential explanations, as well as those that describe different parts of the same causal story, just do not have the potential to be epistemic competitors" (Schupbach 2016, pp. 262–63). Here, both naturalistic explanations of religious beliefs and their theistic explanations seem to be efficient causal explanations, and it is not obvious that they belong to different types of explanation. However, naturalistic explanations of a religious belief and a sophisticated theistic explanation of that religious belief seem to form a causal chain, and hence they "describe different parts of the same causal story". So they do not even have the potential to compete. The upshot of the discussions above is that even if we consider a weaker case of hypothesis competition, the case for a toned-down version of SNE is still hard to make.

While these authors remind us that hypothesis competition is a matter of degree, they also point out that adopting mutual exclusivity as "a simplifying assumption to model scientific reasoning" may lead to the "problem of excluding hypotheses that may well be true" (Schupbach and Glass 2017, pp. 810, 822). For example, when we are searching for the explanations for the mass extinction at the Cretaceous–Paleogene boundary, asteroid impact and massive volcanism are often considered as competing explanations. However, recent research suggests that "the Chicxulub impact accelerated the rate of volcanic activity", and, hence, "there are grounds for questioning the assumption that the impact and volcanic hypotheses are competing" (Schupbach and Glass 2017, p. 823). So the earlier assumption of explanatory competition leads to exclusion of the volcanic explanation, which may well be true as well. It seems that we also need to beware of the problem of excluding theistic explanations, which may well be true, by assuming facilely that they are competing with naturalistic explanations.

## 5. Actual versus Potential Explanations

Nola also makes an important distinction between actual and potential explanations: "An actual explanation is one in which the explanan (i.e., whatever does the explaining) is true and is known to be true and does explain some explanandum (the item to be explained). *A potential explanation is one such that if the explanan were true then it would provide an explanation of the explanandum.* In what follows talk of "explanation" will be taken to be "potential explanation . . . *questions about the evidence for any explanatory hypothesis can be set aside*" (Nola 2013, p. 166, italics mine). Let us agree to this distinction, but I would argue that for his SNE to have actual negative implications for religious beliefs, Nola needs to move beyond the claim that the proposed naturalistic explanations are *merely* potential rather than actual.[17]

The problem is that if a naturalistic explanation is merely a potential explanation, then, even if it is debunking, it only means that *if the naturalistic explanation were to be correct* then it would yield an acceptable explanation of religious beliefs, which presumably has debunking implications.[18] However, it does not follow that all or most religious beliefs would then be rendered unjustified because the naturalistic explanation *may not be in general correct*. Suppose a drug explanation of religious experience is potentially debunking, but, as a matter of fact, most religious experiences are not produced by ingesting drugs. It does not tend to discredit the veridicality of most religious experiences at all. So to determine the force of naturalistic explanations against religious belief, we also need to consider whether it is or can be an *actual* explanation with explanans that are known to be true or probably true. Therefore, questions about the evidence for any naturalistic explanation *cannot* be set aside. Thurow has also considered a case when a proposition X (CSR) would cast doubt on another proposition Y (belief in God). He says, "It is important that belief in X be justified; unjustified beliefs don't have the power to cast doubt on our beliefs" (Thurow 2014, p. 192). Let us raise these questions for the three main explanations proposed by Nola.

## 6. Is Freudianism a Good Actual Naturalistic Explanation?

I agree with Nola that Sigmund Freud's theory of oedipal complexes and the like is dubious, and I am also pretty confident that *some* religious beliefs are indeed formed by wishful thinking. So Freudianism need not be an enemy of religion, but can in fact help religion to purify itself. However, I am not as impressed by Freud's wish-fulfillment explanation *as a general explanation of religious beliefs* as is Nola. In fact, even a researcher on CSR says, "It might be an overstatement to say that among working sociologists, psychologists and anthropologists of religion (especially scientifically minded ones), the theories of Freud and Karl Marx are considered relics rather than serious scientific hypotheses, but it would probably not be far off the mark" (Visala 2011, p. 1). Another scholar thinks that "Freud's theory is now widely discarded" (Van Eyghen 2019, p. 144). In fact, Nola himself points out the limitation of Freudianism: "Freud's theory is not so comprehensive as to debunk all grounds for the belief in God, though it does debunk some" (Nola 2013, p. 172). Here,

Nola seems to claim that the Freudian explanation does *actually* debunk some religious beliefs. Let us critically evaluate this claim.

Freud thinks that belief in God is "born from man's need to make his helplessness tolerable and built up from the material of memories of the helplessness of his own childhood and the childhood of the human race". Of course, these are true of all people, according to Freud. So Freud needs to explain why there are both believers and non-believers but he fails to do so. A contemporary Freudian may say that belief in God is caused by *unresolved* unconscious conflicts, whereas non-believers have successfully resolved those conflicts and, hence, removed the need to believe in God. John Bowker, however, comments: "it would be impossible on psychoanalytic grounds alone to exclude the possibility that God is the source of the sense of God: however much a sense of God may be constructed through . . . the replication of infantile experience, and however much the characterization of God may replicate parental relationships" (quoted in Preus 1987, p. 211). This means that the theist does not need to reject a Freudian account entirely. Freudianism of some form is compatible with a theistic explanation. God may make use of Freudian mechanisms to generate a sense of God.[19] The Freudian mechanism *must* be off-track from the perspective of the naturalistic worldview, so it is no wonder Nola makes this claim confidently. However, it may not be off track in a theistic worldview. So Nola's claim cannot be established independent of the choice of worldviews. In other words, he is begging the question against theists in the formulation of his atheistic argument.

Another reason why different explanations of a single event E may be compatible is that perhaps each of them only provides a *partial* explanation of E. In the context of CSR, David Leech and Aku Visala argue that "it is not true that CSR can provide us with a complete or sufficiently complete causal explanation for belief in God . . . we have good reasons to think that the CSR causal account of religious belief is not one which would allow us to exclude the possibility of God being directly involved in the process". Moreover, "CSR only explains the emergence of religion in a very generic way rather than explaining any particular individual's belief in God" (Leech and Visala 2012, p. 169).[20] Similar points can be made about Freudianism or the MMP-God hypothesis.

Even if theists might accept some form of Freudianism, should they? It depends on the empirical credentials of this theory. Freud's phylogenetic explanations of religion are by now mostly dismissed. Let us consider the implications of his ontogenetic explanations of religion for some God-experients:

(1) These God-experients all have unresolved unconscious conflicts, perhaps in relation to their attitudes to their fathers;
(2) These unresolved conflicts, together with their memories of their childhood and the childhood of human race, cause them to believe in God, and the inner needs, etc., are *projected* onto external reality;
(3) This belief somehow *causes* them to have the theistic beliefs or experiences.

How do we know whether all three claims above are true and whether the religious beliefs or experiences *actually* came about in the alleged way? These are contingent, empirical matters that need to be verified. The Freudian critic needs to adduce reasons for believing that they are all *actually* true, before Freudianism can become a good actual explanation. I do not have the faintest idea how these are to be verified for a particular group of people. They do not seem to be discoverable by either sense experience or introspection. Basically, all claims about the unconscious are actually *theoretical constructs*. So we have to critically assess the empirical credentials of these constructs and the underlying theory of Freud. We can also ask whether the theory can yield predictions about those people? For example, David Hay argues that Freud's theory should lead us to expect that God-experients should be more neurotic than the others. However, the empirical data he collects seem to contradict that: "The data that have been assembled on the psychological well-being of populations reporting religious experience contradict the view that religious experience is associated with poor mental health" (Hay 1990, p. 89).[21]

If we adopt a critical and empirical approach to naturalistic explanations, rather than an a priori approach, William Alston's claims may not be unjustified: "the most prominent theories in the field invoke causal mechanisms that themselves pose thus far insoluble problems of identification and measurement: unconscious psychological processes like repression, identification, regression, and mechanisms of defense; social influences on ideology and attitude formation. It is not surprising that theories like those of Freud, Marx, and Durkheim rest on a slender thread of evidential support and generalize irresponsibly from such evidence as they can muster. Nor do the prospects seem rosy for significant improvement" (Alston 1991, p. 230). In fact, numerous thinkers have already mounted a counter-critique of the critique of Freud.[22]

Another difficulty is that many religious doctrines are psychologically difficult to accept, e.g., selfless love, sacrifice, striving for perfection, and taking up the cross. They are hardly comforting, and this contradicts the Freudian expectation. More importantly, a religious attitude often leads to a radical evaluation of our desires. So "the idea of wish-fulfillment in Feuerbach is a fundamentally wrong turn . . . If a believer confesses his desires before God, this may be with the purpose of seeing their true status in the light of the divine providence. That in turn involves seeing why they . . . need modifying or replacing by more religiously appropriate desires . . . the 'solution' consists in bringing the ego's desires before a standard of what is truly worthwhile . . . and transcending human wishes" (Clarke and Byrne 1993, p. 119).

In fact, projectionism is a doubled-edged sword: we can also use the projection theory to explain atheism (cf. Koster 1989; Vitz 1999). For a more detailed assessment of projectionism, see Kwan (2006).

Therefore, contrary to what Nola claims, Freud's wish-fulfillment theory, despite its relevance for some kinds of religious belief, is not really plausible as a *general* explanation of religious belief. Moreover, it is not clear how it is "open to test", and, as far as some tests can be carried out, the data may not fit Freudianism's predictions.[23] So the claim that "wish-fulfillment is an unreliable BFP" is not really relevant to the evaluation of most cases of theistic belief or experience, because we do not yet have reason to believe that the wish-fulfillment mechanism is really operating on those cases. The conclusion is that we lack good reasons to believe that the Freudian naturalistic explanation is an actual defeater of religious belief or experience in general.

## 7. How Good Is the Cognitive Science of Religion (CSR)?

It is now fashionable for contemporary critics of religion to appeal to CSR, and HADD in particular, to explain religion, since these seem to be more updated scientific research. Critics often give the readers an impression that CSR has already provided exhaustive explanations of religion. However, according to Leech and Visala, this is misleading: CSR "is informative merely about the generic natural underpinnings of human religiosity". However, "just by themselves they don't explain very much" (Leech and Visala 2014, p. 69). Perhaps the "CSR core ideas might face considerable revisions in the future. We are not even sure whether the CSR's main theories, such as the MCI [minimally counter-intuitive] hypothesis . . . are well supported by empirical evidence" (Visala 2011, p. 11). Therefore, "the CSR findings present us with no reason to think that we have a sufficient and exhaustive naturalistic explanation of religion . . . CSR certainly cannot be recruited as evidence that we live in a naturalistic, causally closed universe" (Leech and Visala 2014, p. 71).

Despite all the above reservations, Leech and Visala are inclined to accept the mainstream CSR. However, other scholars are more critical of CSR, including HADD. Powell and Clarke are defenders of the selectionist paradigm, and they mention major criticisms of the selectionist explanations. First, "they are liable to fail to distinguish the current utility of traits from the reason that traits have originated" (Powell and Clarke 2012, p. 465), but "the 'imagistic' religious practices of preliterate societies that are relevant to explaining the origins of religion have social and ecological effects that are very different from

those associated with contemporary 'doctrinal' religions" ([Powell and Clarke 2012](#), p. 466). Second, selectionists are prone "to rely on plausibility alone as a criterion of explanatory acceptability, and in doing so become advocates of speculative 'just so stories.'" They are "content to identify particular uses that an entity or trait happens to have and then proceed as if they have identified *the* functions that have caused those particular entities or traits to come into being and proliferate" ([Powell and Clarke 2012](#), pp. 465–66). "Often, evolutionists use *consistency* with natural selection as the sole criterion and consider their work done when they concoct a plausible story. But plausible stories can always be told. The key to historical research lies in devising criteria to identify proper explanations among the substantial set of plausible pathways to any modern result" ([Gould and Lewontin 1979](#), p. 588). Without such criteria, a speculative just so story may look warranted when it is not.

Powell and Clarke's reply is that the alternative of "byproduct explanations" also "face ... the just so charge" ([Powell and Clarke 2012](#), p. 467). This is because when we say religions are "byproducts of functional cognitive structures ... they necessarily entail appeals to functions, and so they are necessarily vulnerable to the just so charge as are all explanations that involve appeals to functions" ([Powell and Clarke 2012](#), p. 468). For example, "establishing that it would have been evolutionarily advantageous for us to have a HADD is a far cry from establishing that a HADD has actually evolved and that it is disposed to cause religious byproducts ... One cannot reliably infer from a particular adaptive pattern of behaviour (such as hypersensitive agency detection) that there is a specific organ or cognitive mechanism 'designed' by natural selection to produce that behaviour" ([Powell and Clarke 2012](#), p. 469).[24]

Moreover, the byproduct explanation claims that there is a tight causal link between the selected trait and the byproduct. So there is the "need of "demonstrating the causal ontogenetic relation between a mental module and its religious byproduct ... without making a credible case for the conclusion that the operation of the HADD actually causes people to believe in supernatural agents, we are not entitled to this inference ... the lack of evidence for the role of a specialized HADD in generating religious phenotypes is acknowledged by Barrett" ([Powell and Clarke 2012](#), p. 469).

Another difficulty is that to "establish that the HADD causes belief in supernatural agents, we need an explanation of how and why we come to attribute supernatural rather than natural agency ... how and why people continue to believe in the existence of supernatural agents. The HADD hypothesis may explain why we tend to infer agency when we hear rustling in the grass, but it does not account for belief in the ongoing existence of the agents that we (mis)attribute" ([Powell and Clarke 2012](#), p. 469). Since, in the former case, the Type I Error is easy to correct, and the belief in the agent will simply disappear. "In contrast, it seems that attributions of supernatural agents are highly resilient and rarely corrected for." So "we are owed an explanation of why religious byproducts continue to exist over generational time, despite the fact that ... they make no causal contribution to fitness and in fact will often be fitness-reducing". Moreover, since "it is not obviously a *structurally constrained* byproduct ... it is not evident why selection would be unable to produce a functioning HADD that did not misfire so as to lead to persistent belief in supernatural agents" ([Powell and Clarke 2012](#), p. 470). Although all the questions above may not disprove the existence of HADD, they are sufficient to make us aware of the fact that many claims of CSR, including the postulation of HADD, cannot be assumed to be straightforward established facts, especially when they are used to explain religious belief. However, let us assume the existence of HADD, and then evaluate Nola's claim that HADD is obviously unreliable.

## 8. Is HADD a Good Actual Naturalistic Explanation?

When appealing to CSR, Nola mainly focuses on HADD, and emphasizes its unreliability. However, Jong disagrees: "it might be premature to uncritically accept the premise that the cognitive faculties that ostensibly produce religious beliefs are unreliable ... The way we detect agents and infer mentation and how category-based information is triggered

automatically under evidentially ambiguous circumstances were and still are eminently useful tendencies. The hypersensitivity of these cognitive mechanisms is hardly a 'design' flaw; rather, our ability to detect agency and infer psychological states and generate category-based expectations quickly with small, fragmented pieces of information makes normal human activities such as … interacting with strangers and loved ones possible" (Jong 2013, p. 529).

It is important to observe that "hypersensitivity"—jumping to conclusions in the face of logically speaking insufficient data—is a common feature of cognitive mechanisms underlying commonsense or scientific beliefs: "Our everyday beliefs—and indeed, our scientific beliefs—are necessarily under-determined by data, and this is, for the most part, a patently good thing for human life, survival and reproduction … it is perhaps more accurate to think of them as truth-tracking mechanisms that are able to generate useful evidentially underdetermined beliefs ( … about agents' mental states) … these mechanisms are fallible and their reliability in any given context is a matter for empirical investigation" (Jong 2013, p. 529). Consider the infamous 'problem of other minds': "it is … very difficult to verify or falsify these generated beliefs … by ToM [theory of mind] … our belief in others' conscious experiences are not obviously justifiable … it is equally certainly unclear how error-prone they are and whether any given output is a hit or false positive. We might freely assume that all supernatural agents that we detect are false positives, of course, but this would simply beg the question unless there is good reason for such an assumption. It certainly does not follow from the theoretical and empirical research on HADD" (Jong 2013, p. 529).

Adam Green also disputes the uncritical claim that HADD is necessarily unreliable on the whole. First, "there must be some limit to our inclination toward false positives in order for our ability to detect agency to be conducive to survival … [Otherwise] one will be too paralyzed by fear of predators" (Green 2015, p. 63). Second, we need to consider the entire process of agency detection beyond the first moment of the triggering of HADD: consider an example of mannequin. "Suppose, for instance, that, in one's peripheral vision, one sees a human form in a shop window and one has an experience as of a human person … one then … sees that the human form actually belongs to a mannequin … it only takes one a further split second to realize one's mistake" (Green 2015, pp. 67–68). So concerning HADD, "the evolutionary story does not entail diachronic unreliability" (Green 2015, p. 63).

Green and Jong have provided plausible interpretations of HADD that contradict Nola's assertion that HADD is not "delivering truths most of the time". Who is right here? The answer is "nobody knows", because HADD is postulated to exist in our evolutionary past to explain some scenarios. There are many unresolved, and probably unresolvable, questions about its exact nature:

> How sensitive is "hypersensitive"? All faculties will produce false positives. We do not expect 100% reliability. Our vision can also be sometimes too sensitive—we need to anticipate (even when reading). So is it hypersensitive or not?

> How to detect and individuate HADD now?

> How to prove that a certain belief is formed by HADD now?

Visala points out that CSR does not explain "why some single event happened". It only explains "why certain kinds of representations tend to accumulate over time" (Visala 2011, p. 128). So HADD is a psychological mechanism postulated to explain why agency perception (natural and extra-natural agents) emerged in the past. Even if this phylogenetic explanation is correct about the past, we are not sure whether the same mechanism accounts for our agency detection nowadays in individual cases. So is HADD still our current mechanism of detecting agency? Or we are not using HADD but instead another mechanism for detecting agents that is not hypersensitive? If there is another mechanism, then why think we are not using this when we conclude to God? The answers to all these questions seem rather unclear, and it is difficult for Nola to vindicate his picture of HADD over Jong's and Green's.

Nola says, "The very idea of HADD having evolved requires that it be hypersensitive, and this contributes to its unreliability" (Nola 2013, p. 177).[25] However, "contributing to unreliability" does not entail "overall unreliability", which is needed by Nola's argument. The concept of HADD only requires it to be prone to illusions in *some* circumstances, but it is compatible with HADD's having an overall reliability of, say, 60%. If so, then in accordance with reliabilism, the belief produced by this mechanism is still prima facie justified. So if theistic belief is produced by HADD, then this is also prima facie justified. Suppose HADD is only reliable 40% of the time. If HADD is still operative *now* and accounts for our beliefs about agents, then perhaps the theistic belief produced by HADD now would be unjustified, as is our belief in natural agents. If HADD is no longer operative, then it is irrelevant to the justifiedness of our theistic belief *now*. Either way, we do not have defeaters.

### 9. Is MMP-God a Good Actual Naturalistic Explanation?

The major problem for Nola's MMP-God account is that we lack solid evidence to show that his story *actually* happened in our evolutionary past, and the benefits associated with the idea of MMP-God were *actually* the survival advantage that accounted for the emergence of religious belief, not to mention how his account can really explain theistic beliefs *now*. When Hilary Putnam discusses Edelman's evolutionary account of intentionality, he criticizes "the long detour through the at-present totally unproved speculations about our evolutionary history" (Putnam 1992, p. 34). In fact, David Sloan Wilson honestly admits that "the study of culture from an evolutionary perspective is still in the rank speculation stage" (Wilson 2009, p. 320). A neuroscientist points out that stories in evolutionary psychology are "nontestable ... We simply do not know what early humans thought about many ... questions because they left so few artifacts" (Beauregard and O'Leary 2007, p. 208).

Of course, we cannot deny that the CSR theorists have been doing empirical work and making progress in the past decades.[26] I want to make clear that I do not want to discourage empirical research into various kinds of causal influences on religious beliefs. I am inclined to agree with Robert McCauley and Aku Visala's framework of explanatory pluralism, and I have no intention to advocate a hyper-antireductionism which claims that "religious phenomena are, in all interesting respects, *sui generis* or that inquiries about religion *must* be autonomous". For McCauley and Lawson, this is a kind of "special pleading" that is "antiexplanatory, antiscientific, and antitheoretical ... In light of the modern sciences' successes with regard to explanation, prediction, and control over the past 400 years, such hyper-antireductionism is unreasonable and obscurantist" (McCauley and Lawson 2017, p. 5). I also agree with their call for more "collaborative research" between religious studies and CSR (McCauley and Lawson 2017, p. 19), which may be mutually beneficial. However, for CSR explanations to be truly scientific, they also need to be assessed critically by stringent scientific criteria. A positive attitude towards CSR research does not mean that we should make exaggerated claims for its achievements or mistake speculations for established facts.

My main point here is that if Nola is not content with claiming that his proposed naturalistic explanations are *merely* potential explanations, he needs to produce more empirical evidence apart from just-so stories. Although progress is being made and the assessment of the empirical credentials of various CSR theories varies even among the experts, the earlier quotes from various scholars seem to suggest that there may be intrinsic limitations about our knowledge of what *actually* happened in our evolutionary past, which may be difficult, if not impossible, to overcome. At least, even some practitioners admit that "CSR and related fields of research are still in their pre-paradigmatic stage" (Visala 2011, p. 11). Anyway, even if the theory of MMP-God is true, discussions above suggest that it is compatible with an ultimate theistic explanation. If theism is true, would it really be surprising that theistic belief would produce beneficial effects for the society? Not at all.

As for Nola's claim that beliefs that enhance cooperation and survival are presumably not produced by a reliable causal mechanism, again this depends on the prior worldview we accept. Is it really surprising that in a theistic world religion can be true as well as conducive to survival? In fact, Nola is aware of the difficulty to show *all* religious beliefs to be unjustified, because there "may well be causes other than C [causes which are offtrack] which bring about x's believing that p that are untouched . . . Radical debunkers . . . have to show that the kinds of explanation they offer for religious belief are sufficiently all-encompassing that *no* grounds for religious belief survive their critique" (Nola 2013, p. 169, italics mine). Is it really possible to make sure that "*no* grounds for religious belief survive their critique"?

We have also distinguished the phylogeny of religious belief in the human race from the ontogeny of religious belief in an individual. So even if the MMP-God account can explain the phylogeny of religion in the past, it apparently cannot explain contemporary belief in God. Many people come to believe in God on the basis of their experiences of God, not because somehow their beliefs can solve the problem of cheating. In fact, Nola grants that modern secular societies already have means other than religion to solve this problem: "kinds of controls . . . in modern secular societies (e.g., police, courts)". The implication is that the MMP-God can hardly serve as an ontogentic explanation of contemporary belief in God.

As pointed out above, many scholars concede that the mechanisms postulated by CSR are far from providing sufficient explanations of specific religious beliefs, not to mention the widely divergent doctrines of different religions. Even if we admit that those mechanisms contribute to the formation of specific religious beliefs, we can still argue that other factors such as testimony, religious experiences, historical evidence, rational reflection, or natural theology may also contribute to some specific religious beliefs such as monotheistic or Christian beliefs. The implication is that the epistemic status of those religious beliefs cannot be determined from considering the operation of those CSR mechanisms alone.[27] Moreover, we cannot assume the operation of CSR mechanisms has remained substantially unchanged from primeval times to contemporary society. After all, our civilization has significantly evolved and this may lead to refinement of our cognitive mechanisms, including those CSR mechanisms such as HADD, especially due to the incorporation of modern scientific knowledge.[28] Suppose this kind of refinement based on modern science and rational reflection is largely rational, and, for most contemporary educated theists, their religious beliefs are produced by this kind of refined cognitive mechanisms. *If* this is the case, then the naturalistic explanations proposed by Nola alone cannot really provide an adequate explanation of the religious beliefs of contemporary educated theists, not to mention the best explanation. Of course, this is a big "if", and many more discussions and investigations are in order. The point here is that without resolving these issues, it is difficult for us to conclude that his IBE, SNE, is a successful argument.

## 10. Epistemic Significance of Naturalistic Explanations

Nola emphasizes that "HADD . . . is a tendency that can be corrected as we learn more about the world; we replace false attributions of agency with its associated teleological models of explanation by scientific theories . . . Here we are well on the way to establishing the kind of unified approach to science that is found in scientific naturalism" (Nola 2013, p. 178). So a "good case is made . . . for the growing power of naturalistic explanations of religious belief when compared with the much weaker rival folk explanations" (Nola 2013, p. 182).

I have argued that the points made above from the naturalistic perspective, if understood as an argument for naturalism or atheism, are circular or unsuccessful. However, we still need to consider in what other ways naturalistic explanations of religious belief may affect the epistemic status of theism or naturalism. To explore this question, Thurow has used a generalized concept of "casting doubt", which can be analysed in five different ways:

CD1. X entails that Y is false.

CD2. X entails that belief in Y is formed in an irrational way.

CD3. X is evidence against Y.

CD4. X removes/undermines what was once regarded as a source of evidence/good grounds for Y.

CD5. X contributes to explaining various phenomena on the hypothesis that Y is false as least almost as well as the hypothesis that Y is true explains the phenomena. (Thurow 2014, p. 192)

Let X stand for CSR and Y stand for the existence of God. This paper argues that CD1 and CD3 are false in this case. CD2 stands for some kind of debunking argument which is not dealt with in this paper, but Thurow, together with many commentators, would reject that as well. CD5 also seems false here because CSR itself mainly concerns religious belief and has a limited scope of explanation.[29] As for CD3 and CD4, at most they lead to a weak sense of casting doubt: "If CD3 or CD4 were true, X would cast doubt on Y because belief that Y would be justified to a lower degree than it was prior to knowing X. However, this would not entail that Y is unjustified or false; Y might still be quite reasonable" (Thurow 2014, p. 193).

So on the whole, the suggestion that CSR would make a big negative impact on belief in God in all these ways seems to be exaggerated. However, naturalistic explanations may still be epistemically significant because it may affect some putative grounds for belief in God, but the case for this needs to be made on a case-by-case basis. Thurow thinks that "CSR undermines C. S. Lewis' Argument from Desire, but that it does not undermine the cosmological or design arguments". Moreover, "the evidential force of religious experience could be undermined by findings and theories of CSR" (Thurow 2014, pp. 197, 201). However, he adds, "the science of religious experience is far too young to make any useful judgments on this matter" (Thurow 2014, p. 201). I personally disagree with his assessment of the evidential force of religious experiences, especially when we adopt the Critical Trust approach (see Kwan 2011). Unfortunately, we cannot assess all these claims here, but Thurow's discussions show us more nuanced ways to assess the epistemic implications of CSR explanations: we need to turn to more local topics, rather than to draw a global conclusion about the relationship between CSR and religious belief.

I also think that good naturalistic explanations can enhance the coherence of the naturalistic worldview. However, how much it can do so hinges on how well-supported those naturalistic explanations are. I think we need to be more critical of many assumptions of naturalistic explanations and should be somewhat agnostic about evolutionary psychology or CSR's claims about prehistoric times. Of course, we can wait for more empirical evidence to come in. I basically agree with Myron Penner's assessment: "CSR does not provide evidence for atheism, but . . . if one is an atheist, CSR lends 'intellectual aid and comfort'" (Penner 2018, pp. 105–6). At least for theists, they seem to be justified not to go along with Nola's high estimation of the plausibility of the entire naturalistic story and the overall explanatory power of naturalism.

## 11. Conclusions

Nola's suggested naturalistic explanations, as *actual* explanations of religious belief, are far from being proved. We may treat them as potential explanations of religious belief, but I have argued that even if they were all true, they still cannot provide direct support for atheism because they are compatible with sophisticated theistic explanations of religious belief. So I have found Nola's SNE as an IBE against theism wanting. Of course, Nola's second argument, DRB, and the like still need to be thoroughly evaluated. Many scholars have accomplished that and arrived at the conclusion that the DRB is also not proven, while theistic belief has not yet been successfully debunked on the basis of CSR or other naturalistic explanations. This conclusion may or may not be correct, but it suggests that defenders of DRB do have some obstacles to overcome. We may also note that

some other scholars argue that, in fact, findings from modern scientific study of religion are positively coherent with theism (see Barrett 2011; Barrett and Church 2013; Clark 2010). In my earlier book, I have argued extensively that the naturalistic explanations of many kinds of human experience are in fact less satisfactory than the theistic explanations. Interestingly, a defender of a version of DRB, Matthew Braddock, at the same time proposes "An Evidential Argument for Theism from the Cognitive Science of Religion"—the title of his article (Braddock 2018). His main point is that even given the correctness of CSR explanations, their premises are more likely to obtain given a theistic worldview than given a naturalistic worldview. Of course, these are also disputed. For example, Johan De Smedt and Helen de Cruz argue that, while the findings of CSR may cohere with a broadly religious worldview, they seem to contradict the idea of monotheism (De Smedt and de Cruz 2020). Another recent book argues that theistic belief is in fact in tension with the new science of religion (Kvandal 2021). So the debate goes on, and the implications of the new developments in CSR for the rationality of religious belief, no matter positive or negative, are indeed a fascinating topic.

**Funding:** This research received no external funding.

**Informed Consent Statement:** Not applicable.

**Conflicts of Interest:** The author declares no conflict of interest.

## Notes

1    For the general logic or structure of debunking arguments, please see Kahane (2011), Rowland (2019), Egeland (2022), and Klenk (2019). Another relevant paper is McBrayer (2018), which talks about "the epistemology of genealogies".

2    Wilkins and Griffiths' (2013) debunking argument is mainly based on general evolutionary considerations and the lack of the Milvian bridge in the domains of ethics and religion, not founded on the new developments of CSR.

3    For example, see Barrett and Church (2013), Clark (2010), Clark and Rabinowitz (2014), Glass (2016), Jong (2013), Jong and Visala (2014), Launonen (2021), Leech and Visala (2012), Murray (2009), Murray and Schloss (2013), Penner (2018), Stepanenko (2021), Thurow (2013, 2014, 2018), Van Eyghen (2016, 2019, 2020), and Visala (2011).

4    In fact, not all forms of evolutionary debunking arguments are inimical to theism. For example, Plantinga's evolutionary argument against naturalism (EAAN) argues that the combination of metaphysical naturalism with an evolutionary account of our belief mechanisms would produce a general debunking argument against all our beliefs, naturalistic evolution included. Hence, this result would be self-defeating (see Plantinga 2011). However, if we do not assume naturalism but are open to the possibility of theism, then, as pointed out below, the naturalistic evolutionary explanations of our cognitive mechanisms need not be debunked in the end. So Plantinga would insist that the real conflict between evolutionary explanations and the reliability of our cognitive mechanisms occur within the naturalistic worldview but not the theistic worldview. Of course, atheists disagree and suggest that there are ways to solve the problem for naturalists—see Wilkins and Griffiths (2013) and note two. We cannot go into these complicated debates here.

5    Mawson (2014) offers a similar response to this debate.

6    For example, see Swinburne (2004) and Nola (2018). While Swinburne and Nola arrive at rather different conclusions, interestingly, their methodologies and criteria partly overlap.

7    This kind of perspective is consonant with the explanatory pluralism proposed by some major CSR theorists. For example, see chapter one of McCauley and Lawson (2017) and Visala (2014).

8    For an explication of Swinburne's understanding and defense of the Principle of Simplicity, see Swinburne (1997).

9    For example, see Audi (2011, chp. 10), and Sober (2015).

10    Robert Audi suggests that in fact the truth of the Principle of Simplicity itself may be congenial to a theistic explanation: "if the principle of explanatory simplicity is true, that principle itself apparently cannot be non-circularly confirmed empirically and, in any case, might admit of theistic explanation. Arguably, the universe could have been created with governing laws that are in some sense simple and, harmoniously with this, people could have evolved with an intuitive preference for—with a kind of 'mental tool' favoring—simplicity precisely because behavior in accordance with true generalities is more likely to be conducive to survival than behavior in accordance with falsehoods" (Audi 2014, pp. 32–33).

11    Of course, accepting Nola's premises about naturalistic explanation may be difficult, psychologically speaking, for theists who believe that God is in direct contact with them. Daniel Lim thinks that CSR "precludes God from being the *direct* cause of any religious beliefs" (Lim 2016, p. 953), but this is unacceptable for many religious believers. So he suggests that by modeling on a kind of nonreductive physicalism, we can "enable the folk theist to endorse a picture of the world that does not pit supernatural

and natural causes against each other". In his framework, though cognitive scientific mechanisms "are causally sufficient for the formation of religious beliefs," they "do not exclude supernatural agents from also being *directly* causally relevant for the formation of the very same religious beliefs" (Lim 2016, p. 963, italics mine). This may provide some comfort for the folk theist, but I want to point out that the notion of "direct contact" may be illusory, even for our most intimate sense experiences that we feel to be immediate. For example, our visual experience of a table before us, according to the contemporary scientific account, is mediated by the propagation of light in space, the biochemical reactions in our eyes, and a lot of "computations" in our brain.

12  However, we need to admit that adoption of this distinction would generate other problems for religious believers. For example, if we completely reject direct supernaturalistic explanations, then it would be difficult to make sense of Jesus' miracles, answered prayers, and so on. I cannot further explore these problems here, and, in any case, my major claim is only that Nola's argument does not pose a real threat to theism of some sort or another.

13  We can further distinguish the distribution explanation, the explanation of why the originally generated religious beliefs can be distributed and established among the human population, from other explanations. However, I will not go into this in this paper.

14  This point is even recognized by a critic: "Nature is due to Divine agency. And a natural history is an explanation that traces a belief back to natural causes. Thus, a natural history of a belief or set of beliefs is an explanation which excludes the possibility of supernatural intervention in support of those beliefs without excluding the possibility that God is responsible for the whole shebang" (Pidgen 2013, p. 150). Here, Charles Pidgen seems to agree with my distinction of a crude supernaturalistic explanation and sophisticated theistic explanation. In fact, there are other ways to explain the compatibility of a theistic explanation and naturalistic explanations: explanatory over-determination and divine action in a non-deterministic universe (Audi 2014, pp. 31–32). We cannot explore these possibilities here.

15  Earlier explorations of EP include Bechtel (1990) and Bechtel and Richardson (1993).

16  Some advocates of EP lament, "Despite this groundswell of discussion favoring plurality, it has not won much currency in broader theoretical debate in cognitive science" (Dale et al. 2009, p. 3).

17  I am not sure that Nola has always kept this distinction in mind. *Sometimes* he seems to suggest that some naturalistic explanations *are* actual, at least to some extent. For example, he says, "The various scientific accounts recently developed of the causes of religious beliefs have gained considerable explanatory efficacy . . . the naturalistic stance offers the better explanation" (Nola 2013, p. 162).

18  If a naturalistic explanation is an acceptable explanation that has no debunking implication, then it would not pose any threat to theism.

19  For example, Kelly James Clark asks, "Why, after all, could God not have produced in humans a Freudian god-faculty that makes humans universally aware of God under widely realized circumstances? After all, the *sensus divinitatis*, assuming there is one, must have some determinate shape or form . . . To show that there are natural processes that produce religious belief does nothing . . . to discredit it" (Clark 2010, pp. 507–8).

20  While I contend that crude supernaturalistic explanation is not the only type of theistic explanation, I do not rule out *entirely* direct or even miraculous involvement of God in the world. We just need to be very cautious here.

21  So far, I have assumed that the general Freudian theory at least is credible. This, of course, is controversial. Some psychologists, e.g., Hans Eysenck (1985), dismiss Freudianism as psychobabble. Even Adolf Grünbaum, not an unsympathetic commentator, says: "far from having good empirical support, at best these obsessional and oedipal hypotheses have yet to be adequately tested, even prior to their use in a psychology of religion. *A fortiori*, the psychoanalytic ontogeny of theism still lacks evidential warrant" (Grünbaum 1987, p. 166). See also Farrell (1981), Kline (1972), Masson (1990), and Webster (1995)

22  For a response to Feuerbach, see Hans Küng (1990, p. 191ff), Phillips (2001, chp. 4), Marcel Neusch (1982, p. 31ff), and Clarke and Byrne (1993, chp. 5). For a response to Freud, see Küng (1990), Phillips (2001, chp. 8), Neusch (1982, p. 90ff), and Clarke and Byrne (1993, chp. 8). This is just a small sample of the relevant literature.

23  There are a lot of discussions related to this point in the references mentioned in the previous footnote.

24  I agree with both sides' criticisms of the other side. This means that the problem of appealing to just-so stories seems to be a generic problem for the evolutionary explanation of religion.

25  Nola makes another point that HADD has to make a comparative likelihood judgement and "HADD need not always get such likelihood 'judgements' right and so is unreliable in a new way" (Nola 2013, p. 177). However, again this falls short of establishing HADD's *overall* unreliability. He also says, "The cognitive environment in which it operates ought to be sufficiently similar to the one for which it was evolutionarily designed . . . In much earlier non-religious contexts, it is an adaptation that has evolved to function as a detector of specific range of agents (predators), but its later use in religious contexts as an agency postulator, some argue, is not an adaptation but a functionless byproduct of a trait that had evolved earlier to be functional" (Nola 2013, p. 178). Here, it is a just-so story with many details that are simply hard to establish. Who really knows the original cognitive environment in which HADD was evolutionarily "designed"? How can we establish that the extension of HADD to religious contexts is really "a functionless byproduct"? In a theistic worldview, would it not be possible that this extension is part of God's design? So the suggestion that this extension necessarily means even more unreliability seems to beg the question again.

26  For a survey of this kind of empirical research, see Barrett and Burdett (2010) and Barrett (2011).

27    This point has been raised by many scholars, including Thurow (2013).

28    Clark and Rabinowitz (2014) has suggested a process of epistemic winnowing of our belief-forming processes, but they only apply this idea to "*exceptionally long* testimony chains" (p. 122). My idea is more general.

29    Thurow himself thinks that "CSR decreases the degree of justification for belief in God a small degree via way CD5" (Thurow 2014, p. 195, fn 6). I may agree with Thurow here, but I would emphasize that this "small degree" is really slight.

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
