# Peer review of "An Atheistic Argument from Naturalistic Explanations of Religious Belief: A Preliminary Reply to Robert Nola"

_religions, doi:10.3390/rel13111084_

Round 1

Reviewer 1 Report

This paper is a criticism of Robert Nola’s argument for the superiority of naturalistic explanations (NEs) of religious beliefs over religious beliefs themselves. Nola’s position is at least partly influenced by the Cognitive Science of Religion (CSR) literature. Nola presents three NEs which themselves are unrelated to the quest for truth. In other words, not only does Nola view NEs as sufficient to explain religious beliefs, but also as undercutting the validity of said religious beliefs. As Nola writes (quoted in lines 143-4): “there is no reason to suppose that the causal mechanism (i.e., evolution and natural selection) is reliable for the truth of the contained belief … Rather it produces beliefs which enhance cooperation and survival.”

This paper presents a series of counter arguments to Nola’s argument. First this paper points out that NEs are not incompatible with religious beliefs. In other words, even granting the validity of the NEs, Nola’s argument seems to be guilty of overreach. This counter argument not only attacks Nola’s claims, but also forestalls different reactions and counter-counter arguments that Nola might make. I found several of the counter arguments to be persuasive, such as in lines 237-243, 267-276, and 294-297. I also found the counter argument regarding “Actual versus Potential Explanations” in lines 320ff to be effective.

While this counter argument, as presented, is persuasive, the introduction to this section of the paper is decidedly tentative, as it presents the counter argument as a question. It reads: “Are Naturalistic Explanations Incompatible with Theistic Explanations?” Perhaps the author wishes to safeguard his approach against overreach, but it seems the section succeeds at more than merely raising this question.

The “Actual versus Potential Explanations” counter argument provided a good segue into this paper’s next line of attack, which is to question the validity or strength of the NEs themselves. I think this latter section of the paper was also helpful.

Other leading sources from the literature are appropriately cited in the paper.

I note another liability of Nola’s argument (which the author may not wish to pursue); namely, that Nola’s evolutionary approach, and acknowledgement that evolution’s natural processes are unrelated to the quest for truth, compromises epistemology in general and specifically undercuts any claims Nola (or anyone else) makes. As Darwin admitted, “Would anyone trust in the convictions of a monkey's mind?”

Overall the paper is well written, though there are occasional rough spots. For example, line 34 states: “The first argument targets at the alleged existence of God …” This is non sensical. Also, no argument has been defined at this point, so there is nothing that this is referring to. The acronym NE is defined, but not always used. For example, lines 25 and 26-7. Line 295 has a typo (our should be out), and 687 uses an & which should be “and”.

Author Response

Thanks to the constructive comments of the reviewers. I will provide a brief response below, & basically I have revised my essay according to their suggestions.

religions-1973038  reviewer 1

  • Comment: While this counter argument, as presented, is persuasive, the introduction to this section of the paper is decidedly tentative, as it presents the counter argument as a question. It reads: “Are Naturalistic Explanations Incompatible with Theistic Explanations?” Perhaps the author wishes to safeguard his approach against overreach, but it seems the section succeeds at more than merely raising this question.
  • Response: The section heading has been changed to a more positive statement: “Naturalistic Explanations Are Not Incompatible with Theistic Explanations”
  • Comment: line 34 states: “The first argument targets at the alleged existence of God …” This is non sensical. Also, no argument has been defined at this point, so there is nothing that this is referring to.
  • Response: The sentence has been rewritten as below: “The first argument directly argues against the existence of God from the superiority of naturalistic explanations of religious beliefs over supernaturalistic explanations of religious belief (call this kind of argument SNE). This is the argument which will be examined in some details in this paper.”
  • Comment: I note another liability of Nola’s argument (which the author may not wish to pursue); namely, that Nola’s evolutionary approach, and acknowledgement that evolution’s natural processes are unrelated to the quest for truth, compromises epistemology in general and specifically undercuts any claims Nola (or anyone else) makes. As Darwin admitted, “Would anyone trust in the convictions of a monkey's mind?”
  • Response: This is a good suggestion but due to the limitations of space & complexity of the related issues, I cannot deal with this argument in my paper. However, in line with the above suggestion, I have discussed briefly the relevance of this line of argument to our arguments in footnote 4 of the revised paper.
  • Comment: The acronym NE is defined, but not always used. For example, lines 25 and 26-7.
  • Response: The acronym NE has been deleted & replaced by “naturalistic explanation.”
  • Comment: Line 295 has a typo (our should be out), and 687 uses an & which should be “and”.
  • Response: These typos have been corrected. All “&s” have been replaced by “and.”

Reviewer 2 Report

The overall argument of this paper is clear and compelling. The aim is twofold: (1) to argue that naturalistic explanations can be compatible with theism; (2) to argue that the NEs of religious beliefs that Nola prefers are unconvincing. It would help the reader to state this dual aim right in the introduction.

Besides that, the text should be proofread thoroughly. Shortening it will help to gain more focus and perspicuity.

Author Response

Thanks to the constructive comments of the reviewers. I will provide a brief response below, & basically I have revised my essay according to their suggestions.

religions-1973038  reviewer 2

  • Comment: The aim is twofold: (1) to argue that naturalistic explanations can be compatible with theism; (2) to argue that the NEs of religious beliefs that Nola prefers are unconvincing. It would help the reader to state this dual aim right in the introduction.
  • Response: This aim has been added to the paper near the beginning.
  • Comment: the text should be proofread thoroughly.
  • Response: A more thorough proofreading has been conducted. Quite a few mistakes have been corrected, & several improvements in writing have been made.
  • Comment: Shortening it will help to gain more focus and perspicuity.
  • Response: The paper has been shortened a bit.

Reviewer 3 Report

Overall the article is well argued and the author seems to have a good grasp of the literature on CSR arguments. However, I think some improvements and taking into account additional literature are needed before publication.

First, the starting point of the paper is that there is no detailed response to Nola in the literature yet. However, the responses to Nola are nevertheless mostly taken from the existing literature responding to debunking arguments, which seems to indicate that Nola’s argument has been responded to, even if indirectly. Also, some of the articles referenced (such as Launonen 2021) do respond directly to Nola – and also point out that other formulations of the debunking argument are stronger and therefore more deserving of focused attention. Some recent relevant publications are missing, such as Launonen & Visala, Milvian Bridges in Science, Religion and Theology (2022) – note that this article also references some new research skeptical of the hyperactive part of the HADD thesis. The author should make it clearer what the original contribution of the paper is and why Nola’s arguments need an additional response.

Second, the author’s central and repeated objection regarding actual and potential explanations might become clearer if compared with Thurow’s (2014) classification of possible ways CSR theories could undermine religious belief or its grounds. Is possible explanation sufficient for some of these?

Third, the argument regarding levels of explanation would benefit from more analysis in relation to the philosophy of explanation. Currently the up and coming way to describe what the author is getting is “conjunctive explanation” and “explaining away” or “eliminating explanations”. See Jonathan Schupbach, Conjunctive Explanations and Inference to the Best Explanation; as well as David Glass, “Can Evidence for Design be Explained Away”. There is an upcoming volume in the Routledge Science and Religion Series on “Conjunctive Explanations in Science and Religion” – this also includes a chapter by Gijsbert van Der Brink applying the concept to the cognitive science of religion. It would be good for the author to acquire this chapter if possible, so that these portions of the article are not outdated before publication. The idea of "levels of explanation" is also discussed by Visala (2011) with a bit more depth than the author does here.

Fourth, some responses to CSR arguments that would seem relevant to Nola are not cited – why? For example, the Visala and Leech strategy of relying on natural theology and historical reasons to explain and justify religious belief – in that case religious beliefs will not be based merely on universal human tendencies.

Fifth, the author’s point that CSR explanations may not be correct or reliable is interesting and in my judgment correct – there is need to emphasize this point as science and religion work has the tendency to be much less critical of scientific work than scientists themselves are. However, scientists in the field commonly see some portions of historical CSR hypotheses as well established, some as debunked by the progress of science, and still others as needing further work. The author’s argument here would benefit from more detail and nuance like this in order to avoid worries of science skepticism.

I wish the author the best in their future work and hope that we can see this manuscript published eventually.

Author Response

  • Comment: First, the starting point of the paper is that there is no detailed response to Nola in the literature yet. However, the responses to Nola are nevertheless mostly taken from the existing literature responding to debunking arguments, which seems to indicate that Nola’s argument has been responded to, even if indirectly. Also, some of the articles referenced (such as Launonen 2021) do respond directly to Nola – and also point out that other formulations of the debunking argument are stronger and therefore more deserving of focused attention. The author should make it clearer what the original contribution of the paper is and why Nola’s arguments need an additional response.
  • Response:
    • I think as the reviewer points out, the existing response to Nola by far is mostly indirect rather than a direct engagement. Moreover, the response is mainly to Nola’s debunking argument & not to Nola’s argument for atheism. As far as I know, there has been no direct response to Nola’s argument for atheism which involves specific & detailed responses to Nola’s specific arguments (both the IBE framework & specific naturalistic explanations), as I have provided. So this is the original contribution of my paper.
    • The reviewer 3 seems to suggest that Nola’s argument for atheism does not deserve detailed response (at least he wants to press this point against me, perhaps, to make me think harder). I beg to disagree here because while the argument for the incompatibility between naturalistic explanations of religious belief & theistic explanations is sometimes criticized in various works on this issue, this topic has rarely been extensively treated in recent literature. However, some CSR scholars & critics of religion do seem to suggest or imply that there is a kind of incompatibility between naturalistic explanations of religious belief & theistic explanations, & this general impression seems to be shared by many educated people. Nola is an established philosopher of science, & his arguments should be taken seriously. If the alleged incompatibility is an illusion, this is a widely shared illusion which needs to be dispelled more thoroughly. That is why I think a more extended treatment of this argument by Nola is justified.
    • Concerning the debunking arguments against religion, the current responses in journals (at least those in philosophy of religion or religious studies) mainly focus on methodological issues & rarely provide detailed critical assessments of various kinds of naturalistic explanations. In my paper, I combine both treatment of methodological issues & critical assessments of naturalistic explanations & CSR explanations. In fact reviewer 3 appreciates my contribution here: “the author’s point that CSR explanations may not be correct or reliable is interesting and in my judgment correct – there is need to emphasize this point.”
    • His major reservation seems to be that my treatment of the methodological issues needs to be more in-depth & updated. I agree with the need to do so, & has followed his suggestion to improve my paper in this aspect (as shown in the following responses to his methodological points). With my improvements in these (& other) aspects, I think reviewer 3 should be satisfied with my revised paper. This is supported by his expressed “hope that we can see this manuscript published eventually” despite some of his reservations.
    • I have added a paragraph under “Introduction” to explain some of the main points above.

  • Comment: Some recent relevant publications are missing, such as Launonen & Visala, Milvian Bridges in Science, Religion and Theology (2022) – note that this article also references some new research skeptical of the hyperactive part of the HADD thesis.
  • Response: I would really like to read the suggested article, & believe that I would have benefited a lot from reading it. I have tried hard to locate the above reference, which is very new. Unfortunately, I cannot find it in all the university libraries in Hong Kong or on the web. As for the reviewer’s other suggested articles, I have found some & then incorporated their ideas in my revised article. Although including the ideas of this article would certainly improve my article, it should also be pointed out that no single article can include all “recent relevant publications”. It seems to me that my revised paper is already quite long, containing a lot of discussions. So missing this particular one should not be regarded as decisive for my revised article.

  • Comment: Second, the author’s central and repeated objection regarding actual and potential explanations might become clearer if compared with Thurow’s (2014) classification of possible ways CSR theories could undermine religious belief or its grounds. Is possible explanation sufficient for some of these?
  • Response: Thurow (2014) argue that CSR theories could “cast doubt on” religious belief in different ways, some stronger & some weaker. Together with his questions about conjunctive explanations & levels of explanation, these suggest ways to strengthen of my paper’s treatment of the methodological issues. I am thankful for this & have added new paragraphs under the section “The Epistemic Significance of Naturalistic Explanations” to deal with these issues.

  • Comment: There is an upcoming volume in the Routledge Science and Religion Series on “Conjunctive Explanations in Science and Religion” – this also includes a chapter by Gijsbert van Der Brink applying the concept to the cognitive science of religion. It would be good for the author to acquire this chapter if possible.
  • Response: Again, I would really like to read the suggested article, & I have tried hard to locate the above reference, which is “Upcoming.” Unfortunately, I cannot find it in all the university libraries in Hong Kong or on the web. Again, no single article can include all “recent relevant publications” or forthcoming works. So missing this particular one should not be regarded as decisive for my revised article.

  • Comment: Third, the argument regarding levels of explanation would benefit from more analysis in relation to the philosophy of explanation. Currently the up and coming way to describe what the author is getting is “conjunctive explanation” and “explaining away” or “eliminating explanations”. See Jonathan Schupbach, Conjunctive Explanations and Inference to the Best Explanation; as well as David Glass, “Can Evidence for Design be Explained Away”. The idea of "levels of explanation" is also discussed by Visala (2011) with a bit more depth than the author does here.
  • Response: Reviewer 3 suggests the works of Jonah Schupbach & David Glass on Conjunctive Explanations. I welcome this good suggestion. In fact, Glass (2016) has already been cited in my original version. Now I have also made use of the ideas of Schupbach (2016), and Schupbach & Glass (2017) in my new section on methodological issues: “Explanatory Pluralism & Competing Explanations,” which is quite long.

  • Comment: The idea of "levels of explanation" is also discussed by Visala (2011) with a bit more depth than the author does here.
  • Response: Visala (2011) is a book-length defense of explanatory pluralism as the correct approach to the relationship between CSR & religious ideas. Visala (2014) provides a shorter defense of this approach, which is also advocated by McCauley & Lawson (2017)- both of these works have been cited in my original paper (see footnote 7). In my revised paper, I have provided more discussions of explanatory pluralism on the basis of the ideas in both Visala (2014) & McCauley & Lawson (2017). (See the new section on “Explanatory Pluralism & Competing Explanations.”) Although this approach is in general relevant to the assessment of Nola’s argument, I also point out that this alone is not sufficient for a reply to Nola’s argument because their explanatory pluralism mainly deals with the relationship between different levels of science & argues against explanatory fundamentalism or hyper-reductionism. They do not explicitly deal with the tension between the naturalistic worldview & the supernatural worldview at the metaphysical level. So although more in-depth exploration of explanatory pluralism would improve my case somewhat (by providing a more congenial general framework), the essential points in reply to Nola have already been made in my original version.

  • Comment: Fourth, some responses to CSR arguments that would seem relevant to Nola are not cited – why? For example, the Visala and Leech strategy of relying on natural theology and historical reasons to explain and justify religious belief – in that case religious beliefs will not be based merely on universal human tendencies.
  • Response:
  • In fact I have mentioned the “strategy of relying on natural theology” explicitly under the section on “Is MMP-God a Good Actual Naturalistic Explanation?” I write: “As pointed out above, many scholars concede that the mechanisms postulated by CSR are far from providing sufficient explanations of specific religious beliefs, not to mention the widely divergent doctrines of different religions. Even if we admit that those mechanisms contribute to the formation of specific religious beliefs, we can still argue that other factors like testimony, religious experiences, rational reflection or natural theology may also contribute to some specific religious beliefs such as monotheistic or Christian beliefs. The implication is that the epistemic status of those religious beliefs cannot be determined from considering the operation of those CSR mechanisms alone.” The last sentence of the above passage makes exactly the same point contained in the reviewer’s sentence: “in that case religious beliefs will not be based merely on universal human tendencies” So the reviewer 3 seems to overlook this paragraph. (Moreover, earlier in the paper I have also mentioned the relevance of fine-tuning, etc. to Nola’s argument.)
  • However, it is true that I have not mentioned the relevance of “historical reasons”- the words “historical reasons” are now added in the revised version.

  • Comment: Fifth, the author’s point that CSR explanations may not be correct or reliable is interesting and in my judgment correct – there is need to emphasize this point as science and religion work has the tendency to be much less critical of scientific work than scientists themselves are. However, scientists in the field commonly see some portions of historical CSR hypotheses as well established, some as debunked by the progress of science, and still others as needing further work. The author’s argument here would benefit from more detail and nuance like this in order to avoid worries of science skepticism.
  • Response:
    • I think this is a good reminder. I have added a whole paragraph to distance myself from “science skepticism” under the section on “Is MMP-God a Good Actual Naturalistic Explanation?”: “I want to make clear that I do not want to discourage empirical research into variousl kinds of causal influences on religious beliefs. I am inclined to agree with Robert McCaulery & Aku Visala’s framework of explanatory pluralism, I have no intention to advocate a hyper-antireductionism which claims that “religious phenomena are, in all interesting respects, sui generis or that inquiries about religion must be autonomous.” For McCauley & Lawson, this is a kind of “special pleading” which is :antiexplanatory, antiscientific, and antitheoretical… In light of the mdern sciences’ successes with regard to explanation, prediction, and control over the past 400 years, such hyper-antireductionism is unreasonable and obscurantist” (McCauley and Lawson 2017, 5). I also agree with their call for more “collaborative research” between religious studies and CSR (McCauley and Lawson 2017, 19), which may be mutually beneficial. However, for CSR explanations to be truly scientific, they also need to be assessed critically by stringent scientific criteria. A positive attitude towards CSR research does not mean that we should make exaggerated claims for its achievements or mistake speculations for established facts.
    • As for the advice to provide “more detail and nuance,” I would like to include more details about the empirical evidence for various kinds of CSR explanations, but this would make my essay too long. (Another reviewer already has concerns that my essay is a bit too long.) So I refer to such details by referring to some standard works in the footnote 24. I also believe that the added paragraph (see above) sufficiently indicates a more nuanced approach to this issue.